# Tricalcium Phosphate as a Bone Substitute to Treat Massive Acetabular Bone Defects in Hip Revision Surgery: A Systematic Review and Initial Clinical Experience with 11 Cases

**DOI:** 10.3390/jcm12051820

**Published:** 2023-02-24

**Authors:** Matteo Romagnoli, Marco Casali, Marco Zaffagnini, Ilaria Cucurnia, Federico Raggi, Davide Reale, Alberto Grassi, Stefano Zaffagnini

**Affiliations:** 1Ortopedia e Traumatologia Rizzoli Argenta, 44011 Argenta, FE, Italy; 2Clinica Ortopedica e Traumatologica 2, IRCCS Istituto Ortopedico Rizzoli, 40136 Bologna, BO, Italy

**Keywords:** total hip replacement (THR), total hip arthroplasty (THA), revision total hip arthroplasty (rTHA), acetabular revision, articular and periarticular bone loss, bone substitutes, allografts, biomaterials, bone tissue reconstruction, tricalcium phosphate (TCP)

## Abstract

The use of tricalcium phosphate (TCP) as a bone substitute is gaining increasing interest to treat severe acetabular bone defects in revision total hip arthroplasty (rTHA). The aim of this study was to investigate the evidence regarding the efficacy of this material. A systematic review of the literature was performed according to the PRISMA and Cochrane guidelines. The study quality was assessed using the modified Coleman Methodology Score (mCMS) for all studies. A total of eight clinical studies (230 patients) were identified: six on TCP used as biphasic ceramics composed of TCP and hydroxyapatite (HA), and two as pure-phase ceramics consisting of TCP. The literature analysis showed eight retrospective case series, of which only two were comparative studies. The mCMS showed an overall poor methodology (mean score 39.5). While the number of studies and their methodology are still limited, the available evidence suggests safety and overall promising results. A total of 11 cases that underwent rTHA with a pure-phase ceramic presented satisfactory clinical and radiological outcomes at initial short-term follow-up. Further studies at long-term follow-up, involving a larger number of patients, are needed before drawing more definitive conclusions on the potential of TCP for the treatment of patients who undergo rTHA.

## 1. Introduction

The amount of revision procedures following total hip arthroplasty (THA) continues to grow worldwide, with a failure rate of 12% at a 10-year follow-up, mainly due to the increasing average age of first implants and the rising life expectancy of patients [1]. Several challenges could be addressed at the time of revision surgery, and one of the concerns for orthopedic surgeons dealing with THA revision is acetabular bone loss, since it may hamper proper fixation of any revision implant [2]. Different types of strategies and implants for the reconstruction of acetabular bone deficiency have been recently developed. In clinical practice, the most successful techniques combine the use of impaction bone graft (IBG) and allograft [3,4,5,6], hemispheric acetabular component [7,8], cages [9], oblong components [10], iliac screw cups [11], modular acetabular systems or acetabular custom-made implants [12,13]. IBG can be used in combination with other acetabular reconstructive methods. Bone grafts that are commonly used are autologous, allogenic, or synthetic. Autologous bone graft represents the gold standard, providing osteoconductive and osteoinductive properties; however, it is associated with high donor site morbidity and limited availability [14]. Therefore, both allografts and bone substitutes [15] are considered suitable alternatives for bone regeneration.

Synthetic bone substitutes, such as bioactive ceramics, have recently received great focus due to their potential in stimulating cell proliferation, differentiation, and bone tissue regeneration [16]. Among these, tricalcium phosphate (TCP) is one of the most used and effective synthetic bone graft substitutes, [17]. Its solubility is close to that of bone mineral and it is resorbed by osteoclasts [18]. Besides this osteoconductive ability, few data also hypothesize some osteopromotive potential [19]; TCP is completely resorbable thanks to its cell-mediated resorption, and is replaced by newly formed bone when introduced into bony voids, allowing full bone defect regeneration [17]. Furthermore, the mixture of TCP and other compost, such as hydroxyapatite (HA), is gaining increasing interest [20]. In fact, some studies analyzed the ratio TCP/HA due to the influences of solubility of ceramics, and consequently of resorption activity of the two components together [21,22]. Therefore, TCP is emerging as one of the most attractive bone graft substitute materials, used as a pure-phase ceramic or in combination with other components.

The aim of this systematic review was to investigate safety, clinical and radiological outcomes of TCP as a bone substitute to treat severe acetabular bone defects in hip revision surgery.

## 2. Materials and Methods

### 2.1. Search Strategy and Article Selection

A systematic review of the literature was performed on TCP use to reconstruct bone defects in hip revision surgery. This study was registered on the international prospective register of systematic reviews (PROSPERO registration CRD42022370721) [23,24].

A comprehensive literature search was conducted on 29 October 2022 in three electronic databases (PubMed, Embase, and Web of Science), with no time limitation and without any filters, using the following string: ((ceramic bone graft) OR (synthetic bone graft) OR (bone graft substitute) OR (bone substitute)) AND ((revision hip replacement) OR (revision total hip arthroplasty) OR (revision hip arthroplasty) OR (acetabular revision) OR (acetabular defect) OR (acetabular loss)).

### 2.2. Study Selection

According to the Preferred Reporting Items for Systematic Reviews and Meta-Analysis (PRISMA) and Cochrane guidelines [25], the article selection (Figure 1) and data extraction process were conducted separately by two authors (MC and MZ). Since both reviewers agreed on the studies to be included, it was not necessary to involve a third reviewer. The initial title and abstract screenings were made using the following inclusion criteria: clinical studies of any level of evidence, written in English language, and evaluating the use of TCP to treat bone defects in THA. Exclusion criteria consisted of articles that were off-topic, studies written in other languages, literature reviews, preclinical (animal) studies, basic science in vitro articles, case reports, and congress abstracts. Additionally, all references from the selected papers and previously published relevant reviews were also analyzed.

### 2.3. Data Extraction, Outcome Measurement, and Quality Assessment

For the included studies, relevant data were extracted from article texts, tables, and figures, and then summarized and analyzed according to the purpose of the present work. In particular, the following data were collected: year of publication, study design, surgical technique, details of the bone substitute used, survival rate of the implant, number of evaluated patients, patient characteristics, acetabular defects, clinical and radiological follow-up length, clinical and radiological evaluation methods, main results, failures, and adverse events. The survival rate of the revision acetabular cup was summarized including clinical and radiological failures. The efficacy of TCP as a bone graft in THA was evaluated by summarizing data of the clinical scores and data of the radiological examinations, while the safety of the procedures was evaluated by identifying the reported complications.

The quality of the included studies was assessed using the modified Coleman Methodology Score (mCMS) for all studies [26]. The mCMS score ranges from 0 to 100, with a higher score reflecting higher quality. The final score was categorized as excellent (85–100 points), good (70–84 points), fair (55–69 points), and poor (<55 points).

## 3. Results

### 3.1. Article Selection and Studies Characteristics

The electronic search yielded 643 studies. After duplications and non-English articles were removed, 419 studies remained. Of these, 408 were excluded after a review of the abstracts and full-text articles not concerning the use of TCP as bone substitute on the management of acetabular bone loss in revision THA (rTHA). One article was identified through the reference lists. Consequently, 12 articles were selected for eligibility according to the inclusion/exclusion criteria. Four studies were excluded after full-text evaluation: three articles were excluded because they reported the same patients of other included studies updated with a longer follow-up [27,28,29] and, one article investigated TCP in femoral component revisions of THA [30]. Thus, a total of eight clinical studies focusing on TCP as a bone graft to treat acetabular bone defects in hip revision surgery were included in this systematic review.

Among the included articles, the analysis by study type showed eight retrospective case series, of which two were comparatives. Different types of bioactive ceramics including TCP were investigated: six articles investigated biphasic ceramics composed of TCP and HA, and two pure-phase ceramics consisting of TCP.

The evaluation with the mCMS showed an overall poor methodology of the included studies, with an average score of 39.5 points out of 100 (range 28–59).

### 3.2. Patient Characteristics

A total of 231 acetabular revisions were performed on 230 patients affected by THA failures. The surgical indication of rTHA was documented by 5/8 studies, including a total of 175 hips, where aseptic loosening was the diagnosis in all cases. The average age of the patients was 69.9 (65.6/74.3) years, where 65.2% were female and 34.8% were male. The follow-up duration varied from 7 months to 16 years, with an average of 7.7 years. Moreover, acetabular deficiencies were classified according to the system of Parry et al., to the American Academy of Orthopedic Surgeons (AAOS) classification, and to the Paprosky classification [31,32,33]. Finally, biphasic ceramics (TCP + HA) were used in 188 hips, and pure-phase ceramics (TCP) in 43. Details of patient characteristics are reported in Table 1.

### 3.3. Bone Graft Carateristics

TCP was used in different types of bioactive ceramics, constituting of several molecules in different percentages. Four studies utilized a biphasic ceramic composed of 80% TCP and 20% HA (BoneSave^®^, Stryker, UK) [34,35,39,40]; two used pure-phase ceramics consisting of pure beta-TCP (β-TCP) (OSferion^®^, Arthrex, Germany) [38] (Vitoss^®^, Stryker, USA) [41]; one a biphasic ceramic consisting of 40% β-TCP and 60% HA (Bonit matrix^®^, DOT GmbH, Germany); and one study used two biphasic ceramics composed of 45% TCP and 55% (HA) (Eurocer 200+^®^, FH Orthopedics, France), and 35% TCP and 65% HA (Eurocer 400^®^, FH Orthopedics, France) [34,35]. Moreover, four studies mixed biological bone autografts or allografts in combination with bone graft substitutes. Biological grafts utilized were morselized femoral head allografts (two studies), and autologous bone marrow harvests from the iliac crest (two studies). Finally, one article added a collagen–hydroxyapatite fleece (Collapt II^®^, Kyeron, The Netherlands) with osteoconductive properties to the bone substitute [36]. Further details of the bioceramics used are reported in Table 2.

### 3.4. Clinical and Radiological Outcomes

To evaluate clinical outcomes, the Oxford Hip Score (OHS) (four studies) and the Harris Hip Score (HHS) (three studies) were the most commonly used scores. Other scores, like WOMAC, Short Form 36 (SF-36), Short Form (SF-12), Satisfaction Scale for Joint Replacement Arthroplasty (SAPS), Japanese Orthopaedic Association (JOA) score, University of California at Los Angeles (UCLA) activity scale and, Merle d’Aubigné Postel scale, were also used in some studies (Table 3). To evaluate radiological outcomes, graft resorption, migration signs, and radiolucent lines were investigated, according to the method of DeLee and Charnley (seven studies) [42]. Moreover, four articles recorded the grade of heterotopic ossification, according to the system of Brooker et al. [43].

The main finding of the included studies was an overall improvement in functional and radiographic outcomes of patients treated with TCP in the management of acetabular bone loss in rTHA. Radiolucency around revision cups was investigated, where six studies reported a total of 77 (49.3%) patients without any signs of radiolucent lines at the radiographic examination, while 21 (13.4%) patients had signs of radiolucent lines in at least one zone. Absorption, in general, was observed in constant growth throughout the follow-up periods; however, two migrations of the cup were reported. Finally, the survival rate of the implants was reported in five studies with an overall mean survival of 93.3% (Table 3).

The safety of TCP for the treatment of acetabular defects in rTHA was documented by 7/8 studies for a total of 200 rTHAs. No severe adverse events occurred during the surgical procedures. During the follow-up periods, 21 (10.5%) reoperations, not involving the acetabular cup, were performed. Further details of adverse events were reported in Appendix A (Appendix A). Moreover, failure of the implants was documented by seven articles for a total of 200 rTHAs. Four patients (2%) received an acetabular; of these, two were treated for deep infection, one for a migration of the cage, and one underwent acetabular and stem revision, even if the authors did not provide a diagnosis. Further details are reported in Table 3.

## 4. Early Clinical Results of rTHA with a TCP Bone Substitute to Treat Acetabular Bone Loss

### 4.1. Cases Series

In our institute, the bone graft substitute, Cerasorb^®^ Ortho Foam (curasan AG, Kleinostheim, Germany), was used for acetabular bone stock reconstruction in 11 acetabular revisions performed between April 2018 and November 2021. The indication for using Cerasorb^®^ was a massive bone defect of the acetabulum, that required grafting to fill the bone gap. Cerasorb^®^ is a bone substitute composed of a mixture of 85% β-tricalcium phosphate (β-TCP) particles 150–2000 μm in diameter, and 15% porcine collagen consisting of 80% type I collagen, 15% elastin, and 5% type III collagen, with a porosity of 65% [44]. The study group consisted of nine women and two men, with an average age of 68 (range: 46–82 years) years at the time of revision. There was a one-stage revision in six patients for aseptic loosening, and a two-stage revision in two patients for deep infection, where the first stage had been a Girdlestone procedure (one case) and spacer implantation (one case). Finally, three patients had a three-stage revision, the first one being after the position of spacer implantation at the first stage, and the Girdlestone procedure in the second stage, and the other two patients after the failure of the second revision previously performed in another institute.

The diagnosis of periprosthetic joint infection (PJI) was ruled out both preoperatively and intraoperatively. Before surgery, C reactive protein and erythrocyte sedimentation rates were analyzed for every patient, to rule out the presence of active infection. Moreover, in patients suspected of PJI, leukocyte scintigraphy was obtained. During surgery, an analysis of fresh sections of periprosthetic tissue to verify leukocyte counts was carried out. If the pathologist prevented an active infection during the surgery, the prosthetic components were implanted. Meanwhile, during the surgical procedure, permanent histological sections and intraoperative cultures of periprosthetic tissue were collected and sent to microbiology.

Clinical assessments evaluating HHS and radiological evaluation with standard anteroposterior X-rays were performed in all cases, and CT scans were carried out in five cases. The hip center of rotation in standard anteroposterior X-rays of the pelvis was individuated [2]. The distance between the femoral head center and reference line through the teardrop figure was defined as the vertical distance (VD), and the distance between the femoral head center and perpendicular reference line through the teardrop was defined as the horizontal distance (HD) (Figure 2). The changing of those distances was defined as vertical migration (VM) or as horizontal migration (HM) [45].

According to the criteria of Dorr et al. [46], the presence of radiolucent lines, progression of radiolucent lines, radiolucent lines in all three zones, radiolucent lines of 2 mm or wider in any zone, and migration of the acetabular cup to determine the osteointegration and stability of the acetabular prosthetic component at follow-ups, were evaluated. Moreover, as Gross et al. reported [47], the incorporation of the Cerasorb^®^ and the bone allograft was defined radiologically by the presence of trabecular crossing of the graft–host interface. The graft resorption was graded as minor (<1/3 of graft resorbed), moderate (1/3 to 1/2 of graft resorbed), and severe (>1/2 of graft resorbed), and was analyzed using the three zones of the acetabulum defined by DeLee and Charnley [42].

During the surgical procedure, acetabular bone loss was evaluated and classified according to the Paprosky classification [2,33]. Patient characteristics are reported in Table 4. An uncemented acetabular component was implanted in 10 patients: Delta One TT in six cases (LimaCorporate^®^); Delta Revision in two cases (LimaCorporate^®^); and in two cases, the acetabular defect (Paprosky 3B/pelvic discontinuity) proved to be so huge, that a custom-made acetabular implant was required (ProMade, LimaCorporate^®^). In those cases, the CT scan was performed for the development of a custom-made component, and preoperative planning was also carried out for the correct positioning of the prosthetic implant. Meanwhile, a cemented cup Muller (Zimmer^®^) on an acetabular cage (LimaCorporate^®^) was only implanted in one case. Moreover, acetabular screws to achieve further stability were utilized in 10 cases. Finally, in four patients, the femoral component was also revised (LimaCorporate^®^) at the time of index surgery, three due to deep infection and one for aseptic mobilization.

### 4.2. Surgical Technique

All surgeries were performed by the first author (M.R.), who is an experienced specialist hip surgeon. Two different approaches were used during revision surgeries: postero-lateral in 10 patients, and Smith-Petersen in only one patient. The presence of active infection during reimplantation was ruled out in four patients using bacteriologic analysis (cultivation), histologic samples, and an antibiogram. In every surgery, debriding of the granuloma tissue adhering to the acetabulum remaining bone was performed, thus the extent of the acetabular defect was evaluated with the Paprosky classification. In one patient (Paprosky IIIB), it was necessary to stabilize the pelvic discontinuity with the placement of a posterior plate. Subsequently, in nine patients the acetabulum was regularized with hemispheric reamers, and in two cases the acetabulum was prepared with guides for the custom-made implant.

Meanwhile, one frozen, non-irradiated femoral head, from the tissue bank of our institute, was morselized manually with a rongeur, to obtain chips of about 0.5 cm diameter. Afterward, Cerasorb^®^ was used for acetabular bone defect reconstruction (Figure 3), and chips of femoral head allografts were thus positioned above. The bone grafts were impacted with the trial acetabular cup until the stability of the impacted grafts was determined, defined as the presence of a solid wall of impacted bone grafts and without the graft moving under manual pressure. Care was taken to achieve fill of the bone loss with an adequate volume of bone graft substitute.

Standard anteroposterior radiographs of the hip were performed for all patients immediately after the operation.

### 4.3. Postoperative Management

All patients were treated with enoxaparin until full weight-bearing from the first postoperative day. Following the surgery, the hip was placed for one month in a hip brace, fixed at 25° of abduction and in neutral rotation, and unlocked up to 70° of flexion and 10° of extension to allow the flexion exercises, commencing from the first day after surgery. Following an initial touch-down weight-bearing period of two days, partial weight-bearing was allowed for four weeks after surgery, and then progressive weight-bearing, as tolerated, was allowed.

### 4.4. Clinical and Radiological Outcomes

Clinical and radiological examination was performed after the period of one month, six months, 12 months, and then once a year until the last follow-up. At one month of follow-up, a control CT scan in the patients operated with the custom-made cup was performed to verify the correct positioning of the implant, as planned. The mean clinical follow-up period was 2.6 (1–4.7) years and the mean radiological follow-up was 1.7 (4 m–3.6 y) years. Furthermore, one patient was lost at the first radiographic follow-up.

The evaluation of clinical results revealed an increase in pre-operative HHS from an average of 33.5 (15.7–46.2) points to an average of 79.2 (64.7–91.8) points at the most recent follow-up. Moreover, the average preoperative limb length discrepancy was −3.0 (−6.8/+0.3), shrinking remarkably to −0.4 (−3.0/+1.0) after the surgical procedure and remaining constant over the follow-up periods (Table 5).

Radiologically, the HD and VD distances were calculated on the pelvis radiograph after surgery and at the last radiological follow-up. The HD was corrected from preoperative 30.5 mm (17.5–56.9) to postoperative 33.2 mm (21.4–44), the VD was corrected from preoperative 32.2 mm (22.0–90.0) to postoperative 26.7 mm (15.5–69.0). Furthermore, a migration (10 mm of VM; 14 mm of HM) of one acetabular cup 12 months after surgery was observed, but has remained stable in subsequent radiographic controls (Figure 2). The patient was asymptomatic during follow-up periods and was not fit to undergo another revision surgery. Of the 10 cases with radiographic follow-up, no radiolucent lines were observed in five cases, whereas in five patients, a radiolucent line in zone 3, in one a radiolucent line in zone 1, and in two a radiolucency line in zone 2, was observed. Moreover, complete incorporation of the graft was found in nine patients, and partial incorporation was found in two patients (Figure 4). No further migrations in the other patients were observed.

No severe adverse events occurring during the surgical procedure and during the postoperative period were reported. One patient developed a pulmonary emboly after stopping enoxaparin, after six months. There were not any cases of deep infections or prosthetic dislocations of the operated hips. Although a crutch remained necessary in two of the cases, hip pain was nearly or totally alleviated.

## 5. Discussion

The current systematic review revealed the increasing attention on TCP as a bone substitute for the treatment of acetabular bone loss in rTHA. While the number of studies and their methodology is still limited, the available evidence suggests safety and overall satisfactory results of TCP bone grafts in acetabular revision, as confirmed by early case-series reported results.

Particularly, TCP was described in literature since the 1970s [48,49], gaining significant interest as a bone graft substitute due to the development of a promising product to address bone loss, widely investigated by numerous studies derived from maxillofacial and spine surgery research [50,51]. In particular, meta-analyses compared TCP to other grafting materials in treating periodontal defects [52]. TCP research for acetabular reconstruction in rTHA began in the last decade and is still in its dawn, with only eight available clinical studies, of which none are RCTs. A descriptive review tried to summarize bone graft substitutes used in rTHA, suggesting HA and TCP as suitable extenders and potential substitutes when impaction grafting techniques are employed [14]. Regarding radiologic evaluation, Callaghan et al. reported incomplete incorporation of the grafts in CT scans, even when radiographs appear to demonstrate absorption of graft substitute in large acetabular defects [53]. On the other side, Nishii et al. observed considerable variations in rates of β-TCP resorption and new bone formation around femoral prosthesis components through CT evaluations [54]. However, the current literature is characterized by a high heterogeneity of the used products in different anatomic districts, which reduces the strength and evidence of the available reviews. Therefore, TCP has been analyzed for the first time in this systematic review as an acetabular bone substitute, comparing the safety and efficacy as pure-phase ceramics or biphasic ceramics combined with HA. Looking at studies reporting further details on TCP use, TCP was a pure-phase ceramic in two studies [38,41], and a biphasic ceramic in six articles [34,35,36,37,39,40]. Therefore, the TCP/HA ratio of biphasic ceramics changed in the included studies, as well the difference in biological bone grafts used in augmentation, has to be considered. In fact, only four articles utilized biological bone grafts, an autograft was used in two studies [36,39], and an allograft in the other two [34,41]. Moreover, Comba et al. compared TCP used in combination with bone autografts versus biological-only grafts, reporting a lower risk of failure and better clinical and radiographic results, without statistically significant differences [41]. Meanwhile, Hayashi et al. compared TCP, HA, and bulk allografts, reporting a survival rate in the β-TCP group significantly lower than in the HA or bulk allograft groups [38]. Future high-level studies should better investigate the real potential of TCP, also comparing these products directly with other bone grafts. Pooling different TCP products, each documented by sparse data, is not ideal from a methodological point of view and could offer weak results. Thus, while a statistical analysis bears the risk of misleading conclusions, the systematic review allowed for a clear picture of this field.

Among the different aspects of TCP, combination with other compositions or none is one of the most debated. Some preclinical evidence suggests that biphasic ceramics show higher osteogenesis, angiogenesis, and biocompatibility in comparison to the monophasic constitutes, where their bioresorbability could be optimized by varying the TCP/HA content [55]. Various TCP/HA ratios have been evaluated in the literature to determine the best ratio for optimum bone regeneration. However, a direct comparison of the ratio of the constituent phase involved in human clinical trials is still missing, thus there is no general agreement on an ideal ratio for clinical applications [56]. Considering the high heterogeneity of the included studies and the limitations ascribable to a sub-group analysis, future high-level studies should investigate the role of pure or biphasic ceramics, and consequently the TCP/HA ratio, in order to optimize the use of TCP for treating acetabular bone loss in rTHA.

This systematic review showed an overall clinical improvement, even if only three studies reported preoperative values of clinical scores used [36,39,40]. The OHS values achieved in included studies are similar to those reported for a large series of all-revision hips [57]. One study reported a statistically significant improvement in HHS of 48 points from preoperative to final follow-up [40], a result similar to our case-series that achieved a mean increase of 46 points. Moreover, low absorption of the graft was a predictive factor of aseptic loosening and surgery revision [47]. The results of this systematic review are partially in line with the literature. In fact, low incorporation of the graft was observed in three patients who underwent revision of the implant. However, two patients with signs of lysis and low graft absorption did not undergo surgery revision.

The survival rate of the implants in the included studies was calculated using different endpoints. Two articles used the Kaplan–Mayer analysis with radiographic failure as the endpoint, revealing an average of 85.8% (74.2–97.5%) at a mean follow-up of 9.4 years; two studies applied the Kaplan–Mayer analysis with acetabular revision as the endpoint, disclosing an average of 96% (94–96%) at a mean follow-up of 11.12 years, and one article reported implant survivorship of 100% at a mean follow-up of 2.91 years, with acetabular revision as the endpoint. The mean survival rate of the prosthesis in the included studies (93.3%) and in the case-series (90.9%) was similar to those reported in huge registries that analyzed survival rates of rTHA with Kaplan–Mayer curves [58,59], although short follow-up in some of the included studies leaves concerns regarding the reliability of the survival rate of the implants. Further studies with longer follow-ups should show more reliable results on the use of TCP to treat acetabular bone loss in rTHA.

Among the included studies, acetabular bone loss was characterized by a high heterogeneity relative to the different classifications and types of defects. The literature suggests a common consensus regarding lower survivorship of rTHA performed on severe acetabular bone deficiencies compared to those with smaller bone defects [6,60]. In fact, Hayashi et al. also found AAOS type IV acetabular defect to be a significant risk factor for the failure of rTHA [38], and in our case-series, acetabular migration only occurred in a patient classified with a IIIB acetabular defect according to Paprosky.

The limitations of this systematic review reflect the limitations of this field. The literature analysis showed that clinical studies on TCP as a bone substitute in the management of acetabular revisions are few and characterized by a low methodology quality and high heterogeneity. No RCTs were available, and only two comparative studies comparing TC to other bone grafts were conducted. Moreover, there are not enough stratified and homogeneous data based on the type of bioceramic used, making it difficult to merge and compare clinical results, thus impairing the possibility to perform a reliable meta-analysis to draw clear conclusions. Accordingly, it was not appropriate to proceed with the data analysis, and the systematic review offers a state-of-the-art picture of the field. Aligned with this aim, the methodology of the selected studies was evaluated with mCMS, which confirmed the limited quality of the literature. Similarly, the included studies did not always report the exact number and reason for failures, hindering the possibility to obtain a survival curve based on a unique endpoint. Finally, the relatively short follow-up in some of the included studies leaves concerns regarding the durability and effectiveness of TCP as a bone substitute for rTHA.

## 6. Conclusions

TCP for the management of massive bone loss has been proven safe, effective, and associated with a low risk of failure. While the number of studies and their methodology is still limited, the available evidence suggests overall promising results. Consistent with these data, the case-series presented in the current study revealed satisfactory preliminary results. Further studies at long-term follow-up, involving a larger number of patients compared with a control group, are needed before drawing more definitive conclusions on the real potential of TCP for the treatment of patients who undergo rTHA. 

## Figures and Tables

**Figure 1 jcm-12-01820-f001:**
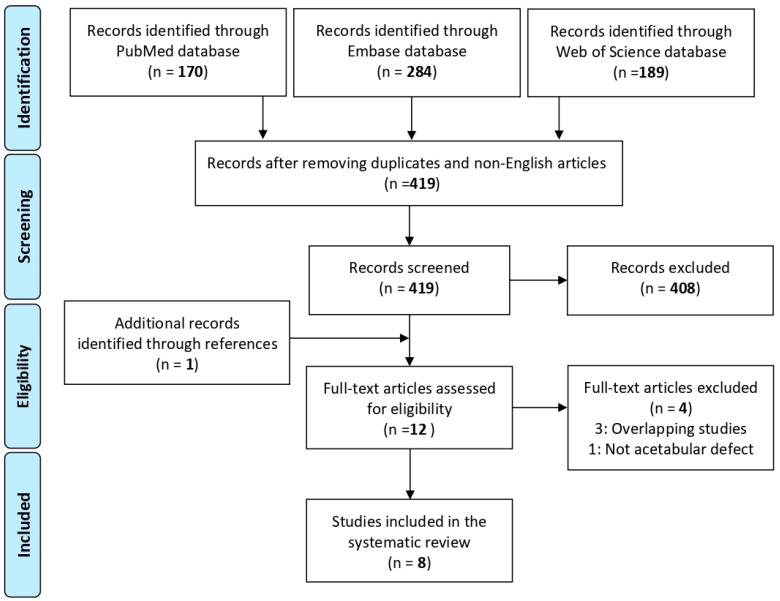
PRISMA (Preferred Reporting Items for Systematic Reviews and Meta-Analyses) flowchart of the study selection process.

**Figure 2 jcm-12-01820-f002:**
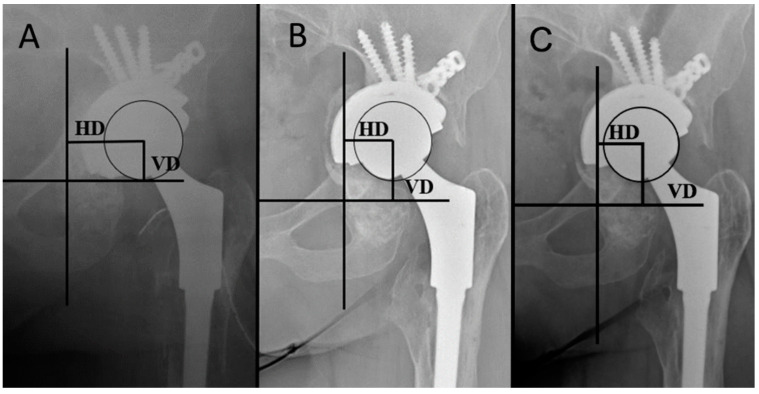
(**A**) Postoperative X-rays showing the method for calculating the HD and the VD; (**B**) Radiological follow-up at 1 year; (**C**) Radiological follow-up at 3.5 years.

**Figure 3 jcm-12-01820-f003:**
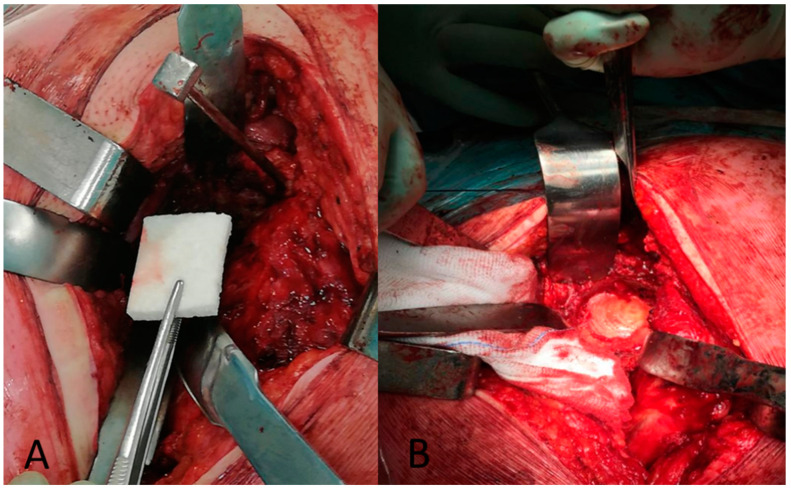
(**A**) Cerasorb^®^ Ortho-Foam; (**B**) The Cerasorb^®^ Ortho-foam placed at the bottom of the acetabulum.

**Figure 4 jcm-12-01820-f004:**
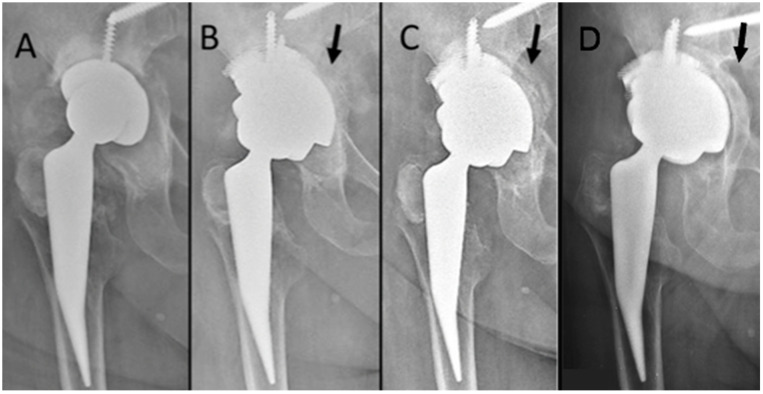
(**A**) Preoperative hip X-ray; (**B**) One-month follow-up X-ray; (**C**) Six-month follow-up X-ray; (**D**) Three-year follow-up X-ray. The black arrow points to progressive bone regeneration.

**Table 1 jcm-12-01820-t001:** Details of patient characteristics.

Authors and Year	Age Mean (Range)	Patients (Sex)	Final F-Up	Bone Graft Substitute	Parry	AAOS				Paprosky				
					B	I	II	III	IV	IIA	IIB	IIC	IIIA	IIIB
Whitehouse et al., 2013 [34]	74 (42–90)	43 (27F/16M)	80 m (69–106)	TCP + HA	33	NA				NA				
Whitehouse et al., 2013 [35]	73 (28–92)	43 (26F/17M)	49 m (SD ± 20)	TCP + HA	33	NA				NA				
Haenle et al., 2013 [36]	74.3 (48–87)	22 (15F/7M)	20.5 m (7–33)	TCP + HA	NA	2	7	13	0	NA				
Schwartz et al., 2015 [37]	68.2 (45–84)	22 (13F/9M)	177.2 m (108–192)	TCP + HA	NA	12	6	8	6	NA				
Hayashi et al., 2017 [38]	66.6 ± 10.4	31 (NR)	105.6 m (SD ± 60)	TCP	NA	0	3	28	0	NA				
		27 (NR)	99.6 m (SD ± 27.6)	HA		0	2	21	4					
		19 (NR)	61.2 m (SD ± 30)	Femoral head allografts		0	1	17	1					
Abdelazim et al., 2020 [39]	65.6 (54–79)	14 (5F/9M)	28.9 m (14.7–34)	TCP + HA	NA	NA				0	0	12	0	2
Gagala et al., 2021 [40]	68.5 (40–83)	43 (18F/25M)	144 m (120–174)	TCP + HA	NA	NA				17	3	3	10	11
Comba et al., 2022 [41]	69 (50–89)	12 (5F/7M)	33 m (12–60)	TCP	NA	NA				5	0	3	0	4
	67 (31–84)	21 (11F/10M)	36 m (13–60)	Femoral head allografts						7	2	1	4	7

AAOS, American Academy of Orthopedic Surgeons; F, female; F-Up, follow-up; HA, hydroxyapatite; m, months; M, male; NA, not applicable; NR, not reported; SD, standard deviation; TCP, tricalcium phosphate.

**Table 2 jcm-12-01820-t002:** Characteristics of bioceramic products used in included studies.

Bioceramic	BoneSave^®^	BonitMatrix^®^	Eurocer 200+^®^	Eurocer 400^®^	OSferion^®^	Vitoss^®^
**Composition**	80% TCP 20% HA	40% β-TCP 60% HA	35 ± 0.5% TCP 65 ± 0.5% HA	45 ± 0.5% TCP 55 ± 0.5% HA	100% β-TCP	100% β-TCP
**Sintering** **temperature**	T > 1200 °C	200 °C	NR	NR	NR	NR
**Crystallinity**	>80%	NR	NR	NR	NR	NR
**Porosity**	50%	60–80%	60%	60–80%	75%	88–92%
**Pore size**	300–500 μm	Micro-and nano-porous range	150–300 µm	300–500 µm	100–400 μm	1–1000 µm
**Granule size**	2–8 mm	0.6 × 0.4 mm	blocks	3–4 mm	NR	100–1000 µm
**Additional** **features**		Embedded in a biologically active silicon dioxide matrix (13%)	Compressive strength 20 Mpa	NR	NR	NR

HA, hydroxyapatite; NR, not reported; T, temperature; TCP, tricalcium phosphate.

**Table 3 jcm-12-01820-t003:** Characteristics of the included studies.

Authors and Year	Study Design	Survival Rate of the Implants	Bone Graft Substitute	Clinical Scale	Score at Baseline and Last F-Up	Failure (Definition and Timing)	Results	mCMS
Haenle et al., 2013 [36]	Retrospective case series	NR	50% BonitMatrix^®^ + 50% autologous BM pelvis	HHS	53 (41–79) 67 (43–93)	No revisions	The use of bone substitutes may gain significance during acetabular revision surgery, partly due to its easy accessibility and broad availability.	36
Whitehouse et al., 2013 [34]	Retrospective case series	94% (CI 99–78) 84 m f-up	50% BoneSave^®^ + 50% femoral head allografts	OHS	NR 31 (NR)	Two acetabular revisions NR (21 m) Deep infection (32 m)	BoneSave^®^ combined with femoral head allograft is associated with low revision rates and high rates of graft incorporation in rTHA.	36
Whitehouse et al., 2013 [35]	Retrospective case series	98% (CI: 85–100) 85 m f-up	100% BoneSave^®^	OHS	NR 36 (6–48)	One acetabular revision Deep infection (16 m)	BoneSave^®^ used without augmentation in rTHA is associated with a low rate of failure and it osseointegrates when suitably loaded by the construct.	28
Schwartz et al., 2015 [37]	Retrospective case series	NR	100% Eurocer 200+ ^®^ or 100% Eurocer 400^®^	OHS	NR 40 (30–48)	No revisions	This study asserts the advantages, safety, and efficiency of the ceramics used in the management of acetabular bone loss in rTHA in a long follow-up.	30
Hayashi et al., 2017 [38]	Retrospective comparative case series	74.2% 105.6 m f-up	100% OSferion^®^	JOA score	NR	NR	The midterm outcomes of rTHA indicate that the type of bone graft and bone defect size may affect the radiographic survival rate when using a KT plate.	59
		81.5% 99.6 m f-up	100% Osteograft^®^					
		94.7% 61.2 m f-up	100% femoral head allografts					
Abdelazim et al., 2020 [39]	Retrospective case series	NR	50% BoneSave^®^ + 50% autologous BM pelvis	OHS	9.5 (42–56) 23.3 (16–30)	No revisions	Dual mobility showed good short-term functional and radiographic outcomes in combination with synthetic bone grafts in rTHA for acetabular defects.	41
Gagala et al., 2021 [40]	Retrospective case series	97.56% 120 m f-up	100% BoneSave^®^	HHS	38.3 (25–55) 86.3 (45–95)	One acetabular revision Aseptic loosening (14 m)	BoneSave^®^ may be suitable for acetabular revision, given that treated patients have better clinical outcomes as compared to the previously cited reports.	56
Comba et al., 2022 [41]	Retrospective comparative case series	100% 35 m f-up	66% Vitoss^®^ + 33% bone grafts	HHS	NR 83 (55–98)	No revisions	β-TCP bone graft substitutes combined with allografts were associated with lower risk of failure compared to biological-only grafts in rTHA.	30
		86% 35 m f-up	100% femoral head allografts		NR 75 (42–96)	Three acetabular revisions Aseptic loosening (NR)		

BM, bone marrow; CI, confidence interval; F-Up, follow-up; HHS, Harris hip score; JOA, Japanese Orthopaedic Association; KT, Kerboull-type; m, months; NR, not reported; OHS, Oxford hip score; rTHA, revision total hip arthroplasty; SD, standard deviation; TCP, tricalcium phosphate.

**Table 4 jcm-12-01820-t004:** Case series.

Patient Characteristics	
Patients (F/M)	11 (9/2)
Mean age (range)	68 y (46–82)
Clinical follow-up	2.65 y (1 y–4.75 y)
Radiological follow-up	1.7 y (4 m–3.6 y)
Paprosky classification	3 type IIC
	3 type IIIA
	5 type IIIB

F, female; m, months; M, male; y, years.

**Table 5 jcm-12-01820-t005:** Clinical and radiological outcomes.

	Preoperative	Postoperative	Final Follow-Up
**Lower limb** **leg discrepancy**	−3.0 (−6.8/+0.3)	−0.4 (−3.0/+1.0)	−0.4 (−3.0/+1.0)
**HHS**	33.5 (15.7–46.2)	NA	79.2 (64.7–91.8)
**HD**	30.5 mm (17.5–56.9)	33.2 mm (21.4–44)	31.9 mm (19.2–44)
**VD**	32.2 mm (22–90)	26.7 mm (15.5–69)	25.7 mm (15.5–69)
**DeLee and Charnley** **radiolucent lines**	NA	NA	2 in zone I
			1 in zone II
			5 in zone III

HD, horizontal distance; HHS, Harris hip score; NA, not applicable; VD, vertical distance.

## Data Availability

Not applicable.

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
