# Peer review of "Tricalcium Phosphate as a Bone Substitute to Treat Massive Acetabular Bone Defects in Hip Revision Surgery: A Systematic Review and Initial Clinical Experience with 11 Cases"

_jcm, 2023, doi:10.3390/jcm12051820_

Round 1

Reviewer 1 Report

The authors state the effectiveness of tricalcium phosphate for revision total hip arthroplasty.

Previous literature is well summarized about clinical and radiographic results using artificial bone. I have two questions about this paper.

1.       In your 11 cases for rTHA, you state six cases were one-stage revision for aseptic loosening, a two-stage revision in two for deep infection and, three patients had a three-stage revision. It is difficult to diagnose infected or non-infected hips.  How did you diagnose infected or non-infected hip? blood examination, imaging, Pathology examination? Please clarify.

2.       In your 11 cases for rTHA, the Paprosky classification was used to evaluate bone defects. In 10 cases, cementless cup was used for reconstruction. Is the cementless cup fixed or in contact with to host bone in all cases? To gain proper fixation cementless cup should be anchored to the host bone without intervening artificial bone. Please clarify.

Reviewer 2 Report

1. Line 37- “one of the most common concerns” : without evidence, stating acetabular bone loss is “most” common concern could be controversial.

2. Line 52- “one of the most used” : awkward phrasing

3. Line 52 – Again, stating TCP as “most used and effective” substitute without any evidence

 -Elaborate more or provide evidence

4. Line 55 – “TPC”? Perhaps typo for TCP? Proofreading needed

5. Line 84 – Inclusion criteria needs to be narrower and stricter. Writing a review paper from articles of “any level of evidence” also makes your paper “any” level of evidence article with low reliability.

6. Line 113 – “full-text articles One article” : No comma? Again, proofreading please

7. Line 112 – Elaborate more on why 408 articles of 419 articles were excluded.

 -You have excluded about 97% of searched article without mentioning definite reason

8. Lines 125~ : “39,5” -> 39.5 All following needs to be changed as well

9. Line 127 : Reason of acetabular revision should be stated as it could be an important factor affecting the result of bone graft (e.g, trauma, aseptic loosening, PJI)

10. Table 1 : For direct comparison, classification of acetabular bone loss needs to be identical.

11. Table 2 : “Extra”? What do you mean by extra?

12. Line 188 – Including your cases, surgical techniques and managements on a “systemic review paper” is irrelevant and unnecessary.

13. Figure 3 – As shown in the figure you provided, tear drop is invisible in cases with acetabular defects due to metal artifact or migration or erosion. Elaborate on how you identified the teardrop in such cases.

14. Table 5 & Lines 322 – If you want to state TCP has “promising results” you need to provide data analysis and P-values, rather than just stating your outcome results in your table.

15. Line 325 – “70s” – Do you mean 1970? 1870? Needs to be written in full term.

Although your study may have aimed to be a systemic review, rather than meta-analysis, your study reviewed various papers which were not uniform in classification, clinical outcome measures. There are way too many variables that could influence the result

Round 2

Reviewer 1 Report

My request is fullfilled.